

# Strain differences between C57Bl/6 and DBA/2 mice (*Mus musculus*) in delayed matching and nonmatching-to-position tasks: impact of sample responses and delay intervals

Kazuhiro Goto

Department of Human Psychology, Sagami Women's University, Sagamihara, Kanagawa, Japan

## ABSTRACT

**Background:** Spatial working memory is commonly assessed in rodents using delayed matching-to-position (DMTP) and delayed nonmatching-to-position (DNMTP) tasks. Although these tasks are widely used to examine memory function, particularly in relation to delay intervals and response requirements, strain differences in task performance remain underexplored. This study investigates spatial working memory in two widely used mouse strains, C57BL/6 and DBA/2, using these tasks.

**Methods:** Mice were trained and tested on the DNMTP task first, followed by the DMTP task. Both tasks were conducted with varying delay intervals and response requirements, allowing for the assessment of spatial working memory across different conditions.

**Results:** Both strains acquired the tasks. However, DBA/2 mice exhibited a smaller decline in accuracy with increasing delay intervals in the DNMTP task compared to C57BL/6 mice. DBA/2 mice also demonstrated more stable performance across both tasks, whereas C57BL/6 mice showed a more pronounced accuracy decline in the DNMTP task than in the DMTP task. In addition, enhancing response requirements during sample trials improved performance in DBA/2 mice for both tasks, a trend that was not observed in C57BL/6 mice. These findings suggest that task-specific variables, such as response modality (*e.g.*, lever pressing *vs.* nose poking) and prior training history, can significantly influence strain performance. Overall, these results emphasize the need for considering strain-specific traits and experimental conditions when interpreting behavioral data, particularly for DBA/2 mice, frequently used as models for hippocampal dysfunction.

# INTRODUCTION

Comparing learning and memory across different mouse strains is essential for understanding these cognitive mechanisms' genetic and neurobiological foundations. C57BL/6 and DBA/2 are among the most extensively studied inbred

Corresponding author
Kazuhiro Goto,
kazuhiro.goto@gmail.com

mouse strains due to their well-documented genetic backgrounds and distinct behavioral phenotypes. These strains consistently display differences in spatial learning and memory (*Paylor, Baskall & Wehner, 1993*; *Ammassari-Teule et al., 1995*). Because spatial learning and memory are highly dependent on the integrity of the hippocampus and its associated neural circuits, examining strain differences can yield valuable insights into the genetic and neurobiological mechanisms underlying cognitive variability (*Moser, Kropff & Moser, 2008*).

Direct comparisons between C57BL/6 and DBA/2 mice have shown that DBA/2 mice typically perform worse than C57BL/6 mice on tasks assessing spatial learning and memory (*Paylor, Baskall & Wehner, 1993*). For example, in the Morris water maze, DBA/2 mice exhibit deficits in learning to locate a hidden platform relative to C57BL/6 mice, often demonstrating longer latencies and a greater reliance on non-spatial strategies, such as swimming along the perimeter (*Owen et al., 1997*; *Upchurch & Wehner, 1988*). Additionally, DBA/2 mice often use less efficient search strategies, such as wall-hugging, indicating difficulty in forming an accurate cognitive map of their environment. In contrast, C57BL/6 mice learn faster and navigate more directly to the platform, reflecting intact hippocampal function. Similar deficits are evident in the radial arm maze, which requires distinguishing between visited and unvisited arms to avoid repeated entries. *Bach et al. (1999)* reported that DBA/2 mice made more re-entry errors into previously visited arms, suggesting deficits in spatial working memory—a process strongly associated with hippocampal integrity.

These observed behavioral deficits in DBA/2 mice align with underlying neurobiological dysfunctions in the hippocampus. Studies have shown that DBA/2 mice have reduced long-term potentiation (LTP) levels, a synaptic mechanism critical for learning and memory (*Jones et al., 2001*; *Nguyen et al., 2000*). Because LTP is considered a critical cellular correlate of memory formation, its reduction in DBA/2 mice suggests impaired synaptic plasticity, potentially contributing to their poorer performance on spatial tasks. Additionally, *Kempermann & Gage (2002)* found that adult DBA/2 mice exhibit lower rates of neurogenesis in the hippocampal dentate gyrus compared to C57BL/6 mice, with these rates correlating with learning performance in the Morris water maze.

Given these documented differences, the present study further focuses on the delayed matching-to-position (DMTP) and delayed nonmatching-to-position (DNMTP) tasks to investigate spatial memory differences between C57BL/6 and DBA/2 mice. The DMTP task requires subjects to memorize a target position (left or right) and match it after a delay, assessing their ability to retain and recall the target position over time. Conversely, the DNMTP task involves remembering a target position and choosing the nonmatching position after a delay. Although the precise cognitive processes differentiating these tasks remain unclear, researchers generally recognize both tasks as spatial working memory assessments (*Pache, Sewell & Spencer, 1999*; *Yhnell, Dunnett & Brooks, 2016*).

Previous research has shown that DMTP and DNMTP tasks reliably measure working memory in C57BL/6 mice (*Goto, Kurashima & Watanabe, 2010*; *Goto et al., 2010*; *Goto & Ito, 2017*). However, direct comparisons using these paradigms between C57BL/6 and DBA/2 mice are limited, and we know little about how DBA/2 mice perform under

standardized conditions in these tasks. I designed this study to determine whether C57BL/6 mice outperform DBA/2 mice in the DMTP and DNMTP tasks and whether the spatial deficits observed in DBA/2 mice in other tasks manifest similarly in these paradigms. Understanding the consistency of strain-specific cognitive profiles across different spatial memory assessments is crucial for validating these inbred strains as models for studying the genetic and neurobiological basis of memory.

## MATERIALS AND METHODS

### Animals

Twelve male mice were obtained from Japan SLC, comprising six C57BL/6J JmsSlc and six DBA/2 CrSlc strains. The use of six mice per strain in this study was based on previous research (*Goto, Kurashima & Watanabe, 2010*), where a sample size of six animals per group provided sufficient data to detect reliable strain differences in similar behavioral tasks. The mice were transported to Sagami Women's University at eight weeks of age and housed in strain-specific cages (29 cm × 19 cm × 13 cm) under a 12:12-h light/dark cycle. Each cage housed six mice of the same strain. Their body weights were monitored and maintained through supplemental feeding (typically 1.75 g per animal daily) and food rewards earned during daily testing sessions. Throughout the study, water was provided *ad libitum* in the housing cages. At the conclusion of the study, all mice were euthanized using $CO_2$ inhalation in accordance with the American Veterinary Medical Association (AVMA) Guidelines for the Euthanasia of Animals. The euthanasia process was conducted in a controlled environment, ensuring minimal distress to the animals. The Animal Care and Use Committee of Sagami Women's University approved this study (No. 2015-06).

### Apparatus

I used two identical operant-conditioning chambers (ENV-307A; Med Associates, Georgia, VT, USA) with internal dimensions of 21.6 cm × 17.8 cm × 12.7 cm. Each chamber was enclosed within a separate sound-attenuating box. The chambers were equipped with three nose-poke response keys (ENV-316M): two on the front panel and one on the back panel. A food well was located at the center of the front panel to dispense 20-mg food pellets (Bioserve, F0071) *via* a pellet dispenser (ENV-203-20). Although a triple LED stimulus light (ENV-322M) was present above the food well, it was not used in this study. A 1.0-A house light was positioned above the back response key. A computer (Optiplex GX620; Dell, Round Rock, TX, USA) interfaced with the chambers controlled and recorded all experimental events and responses.

### Procedure

#### DNMTP training

I initially trained the mice to complete a sequence of four nose-poke responses when the keys were illuminated. Detailed training protocols have been previously reported (*Goto, Kurashima & Watanabe, 2010*). Following this initial training, I trained the mice to perform the delayed nonmatching-to-position (DNMTP) task.

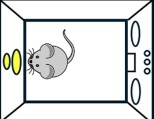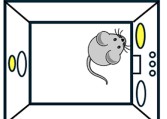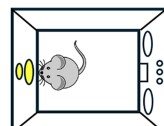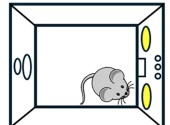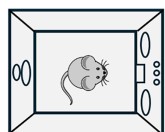

(1) Start:
Houselight and back nosepoke on. Mouse nosepokes to start trial.

(2) Sample:
Left or right nosepoke on. Mouse nosepokes it.

(3) Delay interval:
Back nosepoke on. Mouse nosepokes it.

(4) Choice:
Both nosepokes on. Mouse receives reward when it nosepokes the correct side.

(5) Intertrial interval:
Houselight off for 5-s. Trial begins.

**Trial flow**

**Figure 1 Schematic drawing of the task sequence for the delayed matching-to-position and nonmatching-to-position tasks.** Image source credit: pixabay.

Each trial (illustrated in Fig. 1) commenced with the illumination of the back key. When the mouse nose-poked the back key, the light was extinguished, and one of the two front keys (randomly selected) was illuminated as the sample. Nose-poking the sample key turned off its light, and re-illuminated the back key. Nose-poking the back key extinguished its light, and activated both front keys. A correct response—pressing the key opposite to the sample—resulted in reinforcement. An incorrect response ended the trial without reinforcement and triggered a correction trial, which was excluded from the subsequent analysis. The house light was turned off during the intertrial interval, which was 5 s. The house light remained on for the rest of the session. Each daily session comprised 40 trials, excluding correction trials. The target location was equally distributed, appearing on the left in 20 trials and on the right in 20 trials. Target positions were pseudorandomly assigned across trials. The sessions were terminated if fewer than 40 trials were completed within 45 min. Mice were trained 5 days a week, completing at least 25 sessions (1,000 trials). The training criterion was set as achieving a proportion of correct responses at or above 0.80 in a single session.

*DNMTP test with variable delay intervals*
After completing DNMTP training, I tested the mice with variable delay intervals. In each session, the duration of the back nose-poke presentation was extended to one of five randomly intermixed delays: 1, 2, 4, 8, or 16 s. During the delay, mice were required to respond to the back key to prevent reliance on positional cues for the correct response. Once the delay concluded, the first nose-poke to the back key illuminated both front keys, signaling the choice response phase. Each daily session consisted of 40 trials, excluding correction trials for incorrect responses. Each of the five delays was presented eight times per session, with target locations equally distributed across delays. Trial conditions were systematically counterbalanced to ensure balanced exposure to all variables. The order of trials was pseudorandomized throughout the session to prevent predictable patterns, while maintaining equal representation of all conditions. Sessions were terminated if fewer than 40 trials were completed within 45 min. I conducted 20 sessions, resulting in 800 test trials.

### DNMTP test with variable sample responses

The mice were required to make one, three or six nose-pokes at the sample key in the test sessions examining variable sample responses. The duration of the back nose-poke presentation following the sample response varied, with delays of 1, 4, 8, or 16 s. All other procedures remained unchanged from prior sessions. Each daily session consisted of 48 trials, excluding error correction trials. The target location was equally distributed across trials. Each trial incorporated one of four delay intervals and required one of three sample response levels before progressing. These factors—target location, delay interval, and sample response requirement—were fully combined, resulting in 24 unique conditions, each appearing twice per session. Sessions were terminated if fewer than 48 trials were completed within 60 min. A total of 20 test sessions were conducted, yielding 960 test trials.

### DMTP training

The mice were trained on the DMTP task after completing two blocks of DNMTP test sessions. This task mirrored the DNMTP procedure, with the critical difference that the correct response was the same nose-poke key as the sample. Each daily session comprised 40 trials, excluding correction trials. Sessions were terminated if fewer than 40 trials were completed within 45 min. Mice trained five days per week for a minimum of 35 sessions (1,400 trials) until they achieved a proportion of correct responses criterion of 0.80 or higher in a single session.

### DMTP test with variable delay intervals

After DMTP training, the mice were tested with variable delay intervals using the same procedure as in the DNMTP test. The only difference was that the correct response corresponded to the sample key.

### DMTP test with variable sample responses

Mice also underwent testing with variable sample responses following the same procedures as the DNMTP test, except that the correct response required pressing the same key with a nose-poke as the sample.

### Statistical analysis

The primary outcome measure was the proportion of correct responses. During training, the mean proportion of correct responses was calculated for each subject in 100-trial blocks. To analyze the data, I conducted a two-way mixed design analysis of variance (ANOVA), with strain as the between-subject factor and trial block as the within-subject factor, treating trial block as a repeated measure. For each test phase, the mean proportion of correct responses was calculated across all 20 sessions for each subject. To analyze the data, I conducted a three-way ANOVA, with strain as the between-subject factor and task and subparameter (delay interval and sample response) as the within-subject factors, treating both subparameters as repeated measures. To further examine interactions, I then conducted separate two-way ANOVAs for each task type, with strain as the between-subject factor and subparameter as the within-subject factor, ensuring that
subparameters were treated as repeated measures. Effect sizes were reported using generalized omega-squared ($\omega_G^2$; *Olejnik & Algina, 2003*). Data analysis was conducted using R (Version 4.4.1; *R Core Team, 2024*), and data processing and visualization was performed with the tidyverse package (*Wickham et al., 2019*).

## RESULTS

### DNMTP training

Both C57BL/6 and DBA/2 mice gradually learned the DNMTP task, reaching approximately 0.80 correct responses within 800 trials (Fig. 2A). However, two C57BL/6 mice and one DBA/2 mouse did not meet this criterion within 1,000 trials and required additional training. A two-way ANOVA, with strain as a between-subject factor and trial block as a within-subject factor, confirmed these results. There was a significant main effect of trial block (F(9, 90) = 33.644, $p < 0.001$, $\omega_G^2 = 0.415$), indicating task acquisition over time. No significant effect of strain was found (F(1, 10) = 0.656, $p = 0.437$, $\omega_G^2 = -0.021$), nor was there a significant strain × trial block interaction (F(9, 90) = 1.930, $p = 0.057$, $\omega_G^2 = 0.020$). Although the interaction did not reach significance, the possibility of strain differences in learning rates cannot be ruled out due to the limited sample size in this study.

### DMTP training

DMTP training commenced after completing all the DNMTP tests. Initially, both strains made choices influenced by their prior DNMTP experience; however, they eventually acquired the DMTP task after approximately 1,400 trials (Fig. 2B). C57BL/6 mice achieved the task criterion slightly faster than DBA/2 mice. The first 1,400 trials were analyzed to examine the strain differences during training. A two-way ANOVA with strain as a between-subject factor and trial block as a within-subject factor confirmed the results, showing a significant main effect of trial block (F(13, 117) = 44.581, $p < 0.001$, $\omega_G^2 = 0.644$). There was no significant main effect of strain (F(1, 9) = 3.528, $p = 0.093$, $\omega_G^2 = -0.111$). However, there was a significant strain × trial block interaction (F(13, 117) = 3.038, $p < 0.001$, $\omega_G^2 = 0.078$), indicating the shift from DNMTP to DMTP proceeded relatively more quickly in C57BL/6 mice than in DBA/2 mice. One C57BL/6 mouse did not reach the 0.80 correct response proportion performance criterion. Although this mouse continued training, it finally stopped responding due to a health issue, and its data were excluded from subsequent analyses.

### Interaction of strain, task, and delay intervals

I examined strain and task performance interaction across DNMTP and DMTP tasks. Figure 3A illustrates the proportion of correct responses for each strain at different delay intervals. The proportion of correct responses declined for both strains as the delay interval increased. Notably, C57BL/6 mice showed higher correct response rates in the DMTP task than in the DNMTP task, whereas DBA/2 mice displayed no significant task-dependent differences. This finding suggests that C57BL/6 mice performed better in the DMTP task than in the DNMTP task, whereas DBA/2 mice were less influenced by task type.

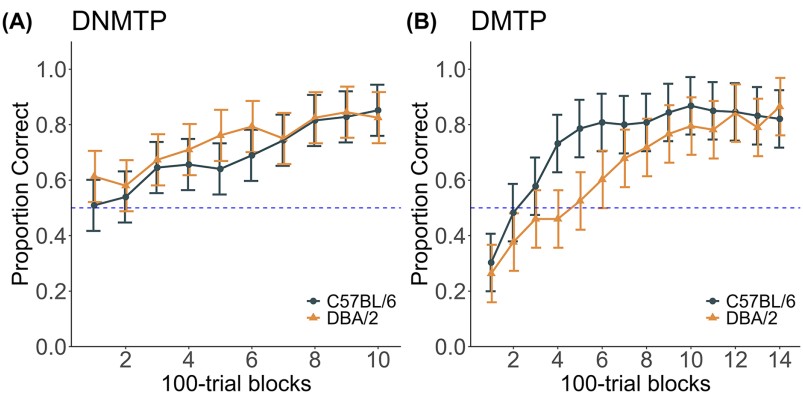

**Figure 2  DNMTP and DMTP Training.** (A) DNMTP training (B) DMTP training. Trials were divided into 100-trial blocks, and the proportion of correct responses was computed for each block.

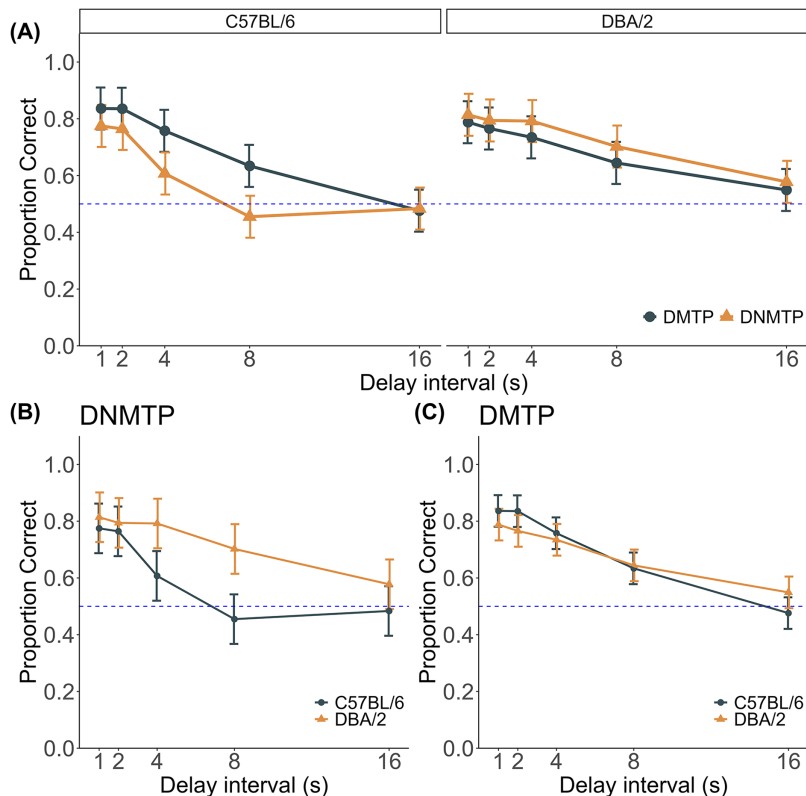

**Figure 3  Proportion of correct responses in the DNMTP and DMTP tests with variable delay intervals.** (A) Task comparison of proportion correct for each strain. (B) Strain comparison of DNMTP performance. (C) Strain comparison of DMTP performance. The proportion of correct responses is shown for each delay interval. Error bars represent 95% confidence intervals adjusted for within-subject error variance (*Loftus & Masson, 1994*). Individual data are shown in Figs. S1 and S2 in the supplementary file.

A three-way ANOVA with strain, task, and delay interval as the main factors revealed a significant strain × task × delay interval interaction ($F(4, 40) = 2.617$, $p = 0.049$, $\omega_G^2 = 0.032$), indicating that the effect of task type on proportion correct varied depending

on the strain and delay intervals. A significant strain × task interaction ($F(1, 10) = 8.012$, $p = 0.018$, $\omega_G^2 = 0.109$) suggested that the performance differences between strains depended on task type. Significant main effects were also observed for strain ($F(2, 10) = 10.913$, $p = 0.008$, $\omega_G^2 = 0.080$) and delay interval ($F(4, 40) = 38.045$, $p < 0.001$, $\omega_G^2 = 0.601$), whereas no significant main effect of task was found.

### DNMTP test with variable delay intervals

Proportion correct declined with increasing delay interval for both strains, but the decline was steeper in C57BL/6 mice than in DBA/2 mice (Fig. 3B). A two-way ANOVA, with strain as a between-subject factor and the delay as a within-subject factor, revealed a significant strain × delay interval interaction ($F(4, 40) = 4.356$, $p = 0.005$, $\omega_G^2 = 0.108$), demonstrating that the effect of delay interval differed between strains. Simple effects analysis for the strain × delay interval interaction indicated that significant strain differences emerged at delay intervals of 4 s or greater ($Fs > 9.895$, $p < 0.01$, $\omega_G^2 > 0.426$). There were significant main effects of strain ($F(4, 40) = 6.601$, $p = 0.028$, $\omega_G^2 = 0.209$) and delay interval ($F(4, 40) = 27.272$, $p < 0.001$, $\omega_G^2 = 0.484$).

### DMTP test with variable delay intervals

Proportion correct declined with increasing delay interval for both strains, but the decline was steeper in C57BL/6 mice than in DBA/2 mice (Fig. 3C). A two-way ANOVA, with strain as a between-subject factor and delay interval as a within-subject factor, revealed a significant strain × delay interval interaction ($F(4, 36) = 3.245$, $p = 0.027$, $\omega_G^2 = 0.09$), indicating that the strains were differently affected by the delay intervals. Simple effects analysis revealed that strain differences occurred explicitly at the 16-second delay interval ($F(1, 9) = 6.946$, $p = 0.027$, $\omega_G^2 = 0.351$), with DBA/2 mice performing better with the most prolonged delay. This finding suggests that extended delays may have less impact on DBA/2 mice than C57BL/6 mice. In addition, a significant main effect of delay interval ($F(4, 36) = 68.027$, $p < 0.001$, $\omega_G^2 = 0.747$), confirming that increased delay intervals reduced accuracy for both strains. However, no significant main effect of strain was observed ($F(4, 36) = 0.085$, $p = 0.777$, $\omega_G^2 = -0.045$), indicating that longer delays adversely affected performance in both strains.

### Interaction of strain, task, and sample responses

Figure 4A depicts the proportion of correct responses for each types of strain under different sample response requirements. Consistent with the findings from Fig. 3A, DBA/2 mice generally showed a higher proportion of correct response than C57BL/6 mice. Unlike the effect of delay intervals, the overall effect of sample response requirements was negligible, but a notable effect was observed in the DNMTP task for DBA/2 mice.

A three-way ANOVA with strain, task, and sample response requirement as main factors revealed a significant strain × task interaction ($F(1, 10) = 14.362$, $p = 0.004$, $\omega_G^2 = 0.093$). Follow-up simple effects analyses for strain × task interaction showed a significant strain difference in the DNMTP task ($F(1, 10) = 14.888$, $p = 0.003$, $\omega_G^2 = 0.527$), whereas no significant difference was observed in the DMTP task ($F(1, 9) = 0.622$,

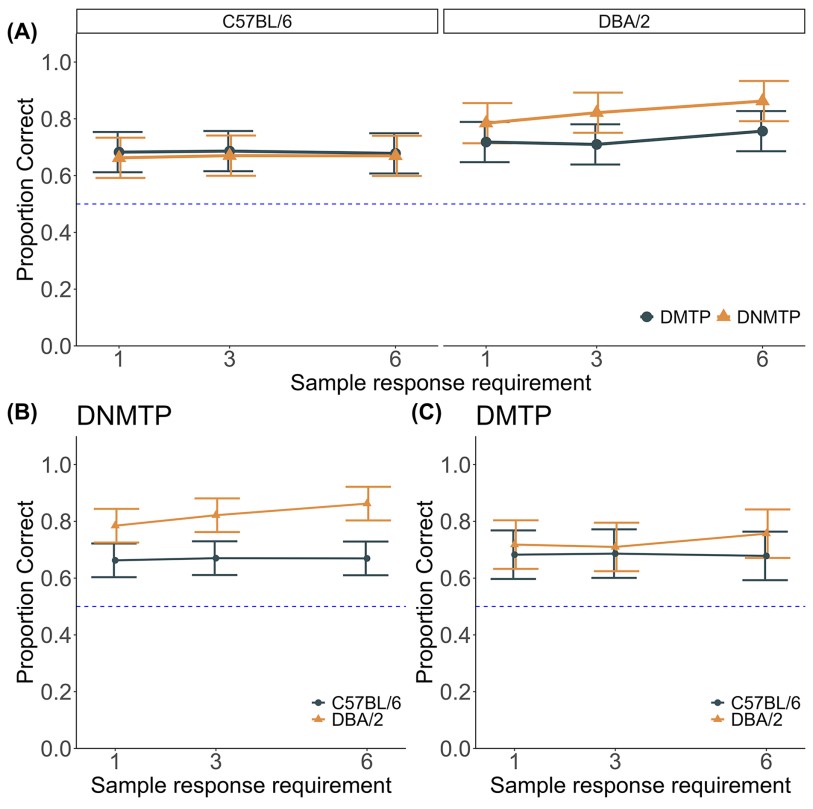

**Figure 4 Proportion of correct responses in the DNMTP and DMTP tests with variable sample responses.** (A) Task comparison of proportion correct for each strain. (B) Strain comparison of DNMTP performance. (C) Strain comparison of DMTP performance. The proportion of correct responses is shown for each sample response requirement. Error bars represent 95% confidence intervals adjusted for within-subject error variance (*Loftus & Masson, 1994*). Individual data are shown in Figs. S3 and S4 in the supplementary file.

$p = 0.451$, $\omega_G^2 = -0.035$). Within-strain comparison indicated that DBA/2 mice showed a significant task difference (F(1, 5) = 22.490, $p = 0.001$, $\omega_G^2 = 0.263$), whereas no significant difference was observed in C57BL/6 mice (F(1, 5) = 0.099, $p = 0.766$, $\omega_G^2 = -0.007$). This suggests that the effect of task type on performance varied between the two strains, rather than being driven by sample response requirements. No significant interactions involving sample response requirements were observed. Significant main effects were found for strain (F(2, 10) = 7.930, $p = 0.018$, $\omega_G^2 = 0.245$) and task (F(1, 10) = 7.608, $p = 0.020$, $\omega_G^2 = 0.048$), whereas no significant main effect was found for sample response requirement (F(2, 20) = 1.325, $p = 0.288$, $\omega_G^2 = 0.005$).

## DNMTP test with variable number of sample responses

Increasing the number of sample responses significantly improved the proportion of correct responses for DBA/2 mice. In contrast, it had a minimal impact on C57BL/6 mice (Fig. 4B). A two-way ANOVA with strain as a between-subject factor and sample response as a within-subject factor revealed a significant strain × sample response interaction (F(2, 20) = 13.458, $p < 0.001$, $\omega_G^2 = 0.037$), indicating that the impact of increasing sample responses differed between the two strains. A significant main effect of strain was observed

($F_{(1, 10)} = 14.371$, $p = 0.004$, $\omega_G^2 = 0.518$), indicating an overall performance difference between the two strains, with DBA/2 mice outperforming C57BL/6 mice. A significant main effect of sample response ($F_{(2, 20)} = 18.807$, $p < 0.001$, $\omega_G^2 = 0.052$) suggested that increasing sample responses influenced task performance, an affect primary driven by DBA/2 mice. *Post hoc* analysis of this interaction showed a significant effect of sample response in DBA/2 mice ($F_{(2, 10)} = 41.514$, $p < 0.001$, $\omega_G^2 = 0.151$), indicating that their performance improved with increasing sample responses. Conversely, no significant effect of sample responses was observed in C57BL/6 mice ($F_{(2, 10)} = 0.295$, $p = 0.751$, $\omega_G^2 = -0.006$).

These results suggest that DBA/2 mice are more responsive to increased sample response, potentially indicating a greater reliance on repetition or additional sample exposure to enhance task accuracy. In contrast, C57BL/6 mice appeared unaffected by the number of sample responses, suggesting that they use a different cognitive strategy that does not benefit from repeated exposure to the sample. The underlying reasons for this difference remain unclear and may involve various interacting factors, such as the physical configuration of the apparatus or differences in locomotor activity and motor perseveration between the strains. I discuss these factors in detail in the Discussion. Further research is necessary to elucidate the mechanisms driving these strain-specific differences.

### DMTP test with variable sample responses

Unlike the results of the DNMTP test, the effect of sample response requirements in the DMTP test was minimal for both strains (Fig. 4C). Nonetheless, increasing the number of sample responses led to a slight improvement in the proportion of correct responses for DBA/2 mice, but this effect was not seen in C57BL/6 mice. A two-way ANOVA with strain as a between-subject factor and sample response as a within-subject factor revealed a significant strain × sample response interaction ($F_{(2, 18)} = 5.441$, $p = 0.014$, $\omega_G^2 = 0.012$), suggesting differential effects of sample responses on performance between the strains. No significant main effect of strain ($F_{(1, 9)} = 0.622$, $p = 0.451$, $\omega_G^2 = -0.035$) or sample response ($F_{(2, 18)} = 2.931$, $p = 0.079$, $\omega_G^2 = 0.005$) was observed, indicating that neither strain nor sample response independently affected performance. However, a simple effects analysis for the interaction revealed a significant effect of sample response in DBA/2 mice ($F_{(2, 10)} = 6.467$, $p = 0.016$, $\omega_G^2 = 0.051$), indicating increased sample responses improved performance. In contrast, no significant effect of sample response was observed in C57BL/6 mice ($F_{(2, 8)} = 0.377$, $p = 0.697$, $\omega_G^2 = -0.001$), showing that sample response requirements did not measurably influence their performance.

## DISCUSSION

This study compared the performance of C57BL/6 and DBA/2 mice on the DMTP and DNMTP tasks. DBA/2 mice outperformed C57BL/6 mice in the DNMTP task, particularly at more prolonged delays and increased sample response requirements. However, no significant strain differences were observed in the DMTP task. These findings contrast with prior studies,

which have typically reported poorer performance by DBA/2 mice on spatial memory tasks such as the Morris water maze or the radial arm maze (*Paylor, Baskall & Wehner, 1993*; *Bach et al., 1999*). This discrepancy suggests that the DMTP and DNMTP tasks may engage cognitive mechanisms distinct from those required by traditional maze tasks.

The differences between these and previous studies' findings likely stem from the unique cognitive demands of the tasks. The Morris water maze, for example, requires mice to navigate and locate a hidden platform using external spatial cues, a process that depends heavily on hippocampal function. Consistent with past research, DBA/2 mice generally perform poorly on such tasks, likely due to hippocampal dysfunction affecting spatial learning and memory (*Owen et al., 1997*; *Upchurch & Wehner, 1988*). In contrast, DMTP and DNMTP tasks involve repetitive sequences that delay intervals may disrupt systematically. These tasks may place less emphasis on hippocampal-dependent spatial processing and emphasize procedural or habitual responses more strongly.

Task structure may also contribute to the observed strain differences. The back key for nose-poking, designed to reset the mice's position during delay intervals, may have inadvertently enabled reliance on positional or external cues (*Paule et al., 1998*; *Panlilio et al., 2011*). This reliance could explain the superior performance of DBA/2 mice, suggesting that these tasks assess working memory and adaptability to task-specific procedural details. Previous research indicates that the ability to switch between cue-based and spatial learning strategies varies by strain, which may help explain why DBA/2 mice performed better in these than in traditional maze tasks (*Cho et al., 2019*; *Schöpke et al., 1991*).

The distinct performances between the DMTP and DNMTP tasks support the idea that these tasks evaluate different cognitive or behavioral processes. The DNMTP task, which requires selecting a nonmatching location, likely demands more significant response inhibition and behavioral flexibility—areas where DBA/2 mice may excel (*Dickson, Calton & Mittleman, 2014*; *Graybeal et al., 2014*). By contrast, the DMTP task emphasizes direct recall of sample locations, potentially explaining why no significant strain differences were detected.

The lack of a sample-response requirement effect in C57BL/6 mice observed in prior studies using lever pressing (*Goto et al., 2010*), raises additional questions. The change in response modality from lever pressing to nose-poking in this study may have contributed to this discrepancy. A study using a touchscreen device found that DBA/2 mice performed worse than C57BL/6 mice in a trial-unique nonmatching-to-position task, even with a shorter delay interval that placed less demand on working memory (*Dickson & Mittleman, 2022*). These results are somewhat inconsistent with those of the present study. Importantly, the observed strain × task interaction suggests that performance difference between C57BL/6 and DBA/2 mice were primarily driven by task structure rather than sample response requirement. DBA/2 mice exhibited greater performance differences between the two tasks than C57BL/6 mice, indicating that procedural variations influenced the strains differently. In addition, transitioning from the DNMTP to the DMTP task could have introduced task-switching costs or interference effects, particularly in C57BL/6 mice (*Graybeal et al., 2014*). These findings underscore the importance of considering

procedural elements, such as response modality and training history when interpreting strain differences.

Although this study provides insight into strain differences in DNMTP and DMTP tasks, several limitations should be acknowledged. First, the small sample size may have reduced statistical power, potentially limiting the ability to detect subtle effects. A larger sample size might have revealed strain differences more clearly. Future studies with larger sample sizes are necessary to confirm these findings and improve generalizability. Second, task order was not counterbalanced across individual subjects due to practical constraints. Counterbalancing would have required a significantly larger sample size to detect order effects with sufficient statistical power. Given this limitation, a fixed task order was used to maintain consistency across subjects and facilitate clearer interpretation of strain differences. However, the fixed order, in which the DNMTP task preceded the DMTP task, may have introduced confounding effects related to task-switching or proactive interference. This order could have particularly affected C57BL/6 mice, given their sensitivity to procedural changes. Future studies should counterbalance task order to better isolate task-specific effects. Addressing these limitations will enhance the interpretability of strain differences in working memory and response flexibility.

## CONCLUSIONS

In conclusion, this study highlights significant strain-specific differences between C57BL/6 and DBA/2 mice in DMTP and DNMTP tasks, with DBA/2 mice consistently outperforming C57BL/6 mice in the DNMTP task under specific conditions. These results suggest that DMTP and DNMTP tasks engage cognitive mechanisms less reliant on hippocampal-dependent memory processes than traditional maze tasks, in which DBA/2 mice often show deficits. DMTP and DNMTP tasks used in this study appear to involve behavioral factors such as motor persistence, locomotor activity, and the strategic use of external cues, in which DBA/2 mice may have an advantage. The absence of a sample-response requirement effect in C57BL/6 mice and the influence of response modality and training history highlights the significance of task-specific procedural elements on performance outcomes. Future research should refine the task designs to isolate better the cognitive processes involved and examine the effects of response modality and task-switching on spatial working memory assessments. Addressing these variables will precisely explain strain-specific cognitive abilities and their underlying neurobiological mechanisms.

## ACKNOWLEDGEMENTS

We acknowledge the use of ChatGPT, an AI language model developed by OpenAI, for English proofreading of this manuscript. The AI-assisted editing focused on improving grammar, clarity, and conciseness, with final approval of the text made by the author. We thank Kaori Uyama for her assistance with animal maintenance and conducting the experiments.

### Funding
This research was supported by a Grant-in-Aid for Scientific Research (B) (23K22371) from Japan's Ministry of Education, Culture, Sports, Science, and Technology. The funders had no role in study design, data collection and analysis, decision to publish, or preparation of the manuscript.

### Grant Disclosures
The following grant information was disclosed by the authors:
Grant-in-Aid for Scientific Research (B): 23K22371.
Japan's Ministry of Education, Culture, Sports, Science, and Technology.

### Competing Interests
The authors declare that they have no competing interests.

### Author Contributions
• Kazuhiro Goto conceived and designed the experiments, performed the experiments, analyzed the data, prepared figures and/or tables, authored or reviewed drafts of the article, and approved the final draft.

### Animal Ethics
The following information was supplied relating to ethical approvals (*i.e.*, approving body and any reference numbers):
The Animal Care and Use Committee of Sagami Women's University approved this study.

### Data Availability
The data is available at OSF: Goto, Kazuhiro. 2025. "Strain Difference between C57BL/6 and DBA/2 on DMTP and DNMTP Tasks." OSF. February 14. doi: 10.17605/OSF.IO/B9A6T.

### Supplemental Information
Supplemental information for this article can be found online at http://dx.doi.org/10.7717/peerj.19200#supplemental-information.

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
