# Peer review of "Strain differences between C57Bl/6 and DBA/2 mice (Mus musculus) in delayed matching and nonmatching-to-position tasks: impact of sample responses and delay intervals"

_PeerJ, doi:10.7717/peerj.19200_

## Round 0.1 · original submission · Minor Revisions

I apologize for the length of time taken to reach a decision on your manuscript. Over the holidays, it was especially challenging to find reviewers and I have decided to proceed with a single review and a thorough read of your manuscript myself. The reviewer was very positive about your work and had only a comment about the reporting of results approaching significance. I agree with the reviewer’s concern here. I have additional comments of my own. I think these can be addressed with a minor revision.

Please avoid using since and while in non-temporal contexts (e.g., line 44, check throughout).
Please provide some references at the end of line 77.
Delete “our” on line 91.
It is fine to write in first person. Replace “the author” with “I”
Line 133, could this criterion be reached in a single session then? Just clarify please.
Please describe the trial configuration exactly for each phase. For example, each of the delays were presented 8 times in a session in random order in the DNMTP Test with variable delays, correct? Were these presented in blocks or completely random?
For the Variable Sample Response phase, I was not clear what this meant. Also please indicate the composition of the 48 trials. Why 8 more trials in this phase?
Why did you train all of the mice first on DNMTP?
Please be clear that you included the interaction of strain and interval in your ANOVA.
It isn’t clear to me how you broke down the significant 3-way interactions. It would make sense to discuss this omnibus ANOVA first, and then split by task to determine whether there are strain by delay interactions for each task (which you had already reported above). Then, if there are significant two-way interactions, analyze whether there are strain differences at each delay, as you also reported above.
Please include a limitations section where you address the small sample size and confounded test order.
Please integrate more references from other researchers into your discussion.

·

Basic reporting

This manuscript was exceptionally well written. The author acknowledges the use of use of ChatGPT for
English proofreading of this manuscript focusing on improving grammar, clarity, and conciseness. The true scholarship comes from the experiment and interpretation, which are entirely the author's own. As such this appears to be a strong and appropriate use of AI to improve the presentation, and I have no concerns.

Experimental design

This study was well designed and executed. The sample size is modest, but given that the same animals are trained and tested over a very long time having a large sample would not be practical. The statistic were expertly conducted and reported.

Validity of the findings

My only concern in this section is that the author may be overly cautious in avoiding a type 1 error and in a couple places might have committed a type 2 error. For instance, line 190-192. The author correctly states "nor was there a significant strain × trial block interaction (F(9, 90) = 1.930, p = .057, ËG² = .020)." With p>0.05 this is not significant. However, the next sentence says: "These results suggest that both strains learned at similar rates." which is probably not correct. They didn't have a significant difference in their learning rates, but that is different from "similar rates". It is likely that they are different in their learning rates, but in this case the test was underpowered.

Additional comments

Overall this is a well designed and executed study. The inclusion of a range of tests and the different strains make this a very useful study highlighting that researchers must be careful in choosing their behavioral tests, and that the choice of strain can also have a large influence on the outcomes.

---

## Round 0.2 · Minor Revisions

Thank you for a straightforward contribution to the literature. I just have a few minor corrections/clarifications to request before I can formally accept the MS.
Please note that line 234 is missing the word "interaction" after "interval."
Line 219, please present analyses to probe this interaction. It would be helpful to plot the significant interactions.
Line 236, do you mean you conducted separate ANOVAs by task type and then examined the 2-way interactions between strain and delay interval? This is what you indicated in the data analysis section but is not described here. Then I see you two have subsections by task type but these headings are at the same level of heading as the omnibus ANOVA so it is still a bit unclear. The same comment applies for the subsequent analyses.
Line 247, you could delete "Follow-up analyses indicated that.." and just start the sentence with "Simple effects.."
I would caution you about interpreting your effects of trial block as indicating changes over time (line 204) as you have treated trial block like a categorical variable, not as a continuous linear trend. Therefore, you really need to do look at post-hoc tests to see where the significant differences lie. At least refer to a figure to support this interpretation.

---

## Round 0.3 · Minor Revisions

I appreciate the revisions but I don’t think the statistics are presented appropriately yet. You say that you did NOT conduct separate ANOVAs by task to explore the three-way interactions, but you still indicate that this is what you plan to do on lines 191-192. Indeed, it does appear to be what you did unless I am confused about the sections under headings, “DMTP Test with variable delay intervals” and the following section. You have to be consistent with your analysis subsection. Furthermore, it is the appropriate step to take to analyze the interaction.

Line 241, this doesn’t make sense – what do you mean you analyzed the two-way within the same three-way model?

Line 221, you don’t need “and” here. You do need to describe which blocks significantly differed from the others.

Lines 224-225, I don’t think you can make this statement as you also have significant strain differences in blocks 2 and 4 and you don’t present any evidence to suggest the differences were larger in blocks 5 and 6. Also, you don’t have significant effects in blocks 7 and 8. I think you need to treat trial as a continuous (or at least ordinal) variable rather than analyzing blocks as a discrete factor and allow it to interact with strain so that you can test whether there is a linear trend of an increase in performance and if this differs depending on group. But you would have to adopt a GLMM approach to account for the trial by trial data.

Line 284, how did you break down the strain by task interaction here? I don’t see two separate analyses of task based on each strain or of strain based on task. You need to choose one of those approaches and conduct these follow up tests or report simple main effects as you do on line 326.

Your discussion will likely change based on revisions to your analyses.

---

## Round 0.4 · accepted · Accept

Thank you for your patience in making these last revisions to the results. I still think that the effect of trial block should be treated differently as I think you are looking for a linear effect like a change over time, not interested in looking at group differences at particular blocks. But given that the effect of block was secondary to the importance of the other variables, I will let it stand as you have reported it.